


# Application of Gamma functions to the determination of unit hydrographs

Hongyan Li [1,*], Yangzong Cidan [2]

*[1] Key Laboratory of Groundwater Resources and Environment, Ministry of Education, Jilin University, Changchun, 130021,China; lihongyan@jlu.edu.cn;*
*[2] Key Laboratory of Groundwater Resources and Environment, Ministry of Education, Jilin University, Changchun, 130021,China;aerlin321@163.com*

*\* Correspondence: lihongyan@jlu.edu.cn*

**Abstract:** There are many methods for calculating unit hydrograph, such as analysis method, trial algorithm and least squares method. But these methods have certain requirements for flood datas and the unit hydrograph may not be optimal. Based on the theory of composition, a hydrological system was viewed as a generalized collection in this study and Gamma functions were used to simulate the basin convergence process. At the same time, the Gamma function is parameterized and the parameters of Gamma function are optimized by genetical gorithm, which is based on the minimum error between the calculation of confluence process and the measurement process, before deriving the unit hydrograph. The Collins iteration method was used to compute the unit hydrograph. The results of actual calculated examples showe that this method is more precise than other methods, while it can also illustrate the law of runoff.

**Keywords:** Theory of composition; Gamma function; Genetic algorithm; Unit hydrograph; Collins iteration method

## 0. Introduction

The unit hydrograph is an important method for simulating the flow concentration of a conceptual hydrological model; it was proposed by Sherman (1932). In actual applications, the derivation of the unit hydrograph is still an important component when forecasting the basin rainfall and runoff.

A unit hydrograph ( Viessman, 1989; Raghunath, 2006) refers to the unit net constant rainfall uniformly distributed over a watershed of unit surface and for unit duration. Periods of 1, 3, 6, and 12 h can be selected and the unit rainfall (runoff depth) is generally 10 mm. The actual net rainfall often does not equal 1 unit for these time periods, so it is necessary to make two basic assumptions when calculating the watershed flow concentration process.

(a) Assumption of linear hydrological system: if the unit period rainfall is k units, the formation of the flow process is k times the unit hydrograph ordinate.

(b) Superposition assumption: if the rainfall lasts m periods, the formation of the flow process is the superposition of each rainfall period.

Based on the above assumptions, the basin outlet section discharge hydrograph can be expressed as:



$$Q_i = \sum_{j=1}^{m} \frac{h_j}{10} q_{i-j+1} \quad \begin{cases} i = 1,2, \ldots, l \\ j = 1,2, \ldots, m \\ i - j + 1 = 1,2, \ldots, n \end{cases} \quad (1)$$

where $Q_i$ is the basin outlet section of each period discharge in m3/s;
$h_j$ is the rainfall in each period in mm;
$q_{i-j+1}$ is the ordinate of the unit hydrograph in each period in m3/s;
$i$ is the number of periods for the basin outlet section flow hydrograph;
$j$ is the net number of rainfall periods;and
In essence, the unit hydrograph is the characteristic of watershed concentration in the form of
discharge hydrograph, i.e. concentration curve (d.johnstone, 1949). The calculation method of unit
hydrograph confluence takes the basin as a whole and assumes that the net rain is uniformly
distributed over the whole basin, without considering the inhomogeneity within the system; the
basin confluence system is a linear time-invariant system, and at the same time, it is viewed that
the net rainfall and the formation of the flow process are in agreement to superposition
relationship. Therefore, the essential characteristics of the unit hydrograph are lumped resistance,
linearity, and time invariance.
Conceptually, the unit hydrograph is a linear time-invariant basin system with a convergent
flow curve. However, the physical mechanism of the watershed conflux is not considered by the
method used to derive the unit hydrograph (Ramirez, 2000). The principle used to calculate the
unit hydrograph is based on the system input (rainfall), which is converted using the unit
hydrograph to determine the system response output (outlet section flow process), where the error
is minimized. The traditional methods of derivation are as follows.
Analytical approach: the basin outlet section of the surface runoff is $Q_1, Q_2, \ldots, Q_l$, the
rainfall process is $h_1, h_2, \ldots, h_m$, where Eqn. (1) comprises $q_1, q_2, \ldots, q_n$ unknown linear
algebraic equations. The solutions of the equations can be obtained using a unit hydrograph.

$$q_i = \frac{Q_i - \sum \overline{\frac{h_j q_{i-j+1}}{10}}}{\frac{h_1}{10}} \quad \begin{cases} i = 1,2, \ldots, n \\ j = 1,2, \ldots, m \end{cases}$$

60                      (2)

where n is the number of periods of the unit hydrograph and $n = l - m + 1$.
In theory, there are no errors with the analytical approach when using the rainfall runoff
measurements. This approach can obtain the correct answer if the watershed conflux conforms to a



linear time-invariant system (when the convergence of watershed system meets multiple
proportions and superposition assumption, for a linear system). However, (1) the data
measurements have errors and, (2) rainfall-runoff system is not a linear time-invariant system. Due
to the accumulation of errors, unreasonable solutions often occur. For example, the unit
hydrograph may be irregular or appear to be negative.
(b) Trial and error: the unit hydrograph is assumed to be $q_i^{'}$. The flow $Q^{'}$ determined by
the unit hydrograph is then compared with the measured flow $Q$. The unit hydrograph $q_i^{'}$ is
produced when the error between Q' and Q satisfies certain error.
The trial and error method proposed by Collins (Collins, 1939) uses an iterative strategy. For
a period of uneven net rainfall, the computational convergence is fast if the time period is large.
However, the Collins iterative method has the following disadvantages: (1) the iterations are based
on the initial unit hydrograph, but there is no strict method for selecting the initial unit hydrograph;
and (2) trial-error approach does not necessarily converge to a solution.
(c) Least squares method: the measured surface runoff is assumed to be $Q$ and the error is
as follows: $\varepsilon = Q - Q^{'}$. If $\sum \varepsilon^{2} = \sum (Q - Q^{'})^{2} \rightarrow \min$, we try to convert Eqn. (1) into a
normal equations system where the number of equations is equal to the variables. The optimal
estimation of $q_i$ can be solved using the least squares method. The theory of the least squares
method is better, but the results are sometimes fluctuating or negative.
There are many methods for determining unit hydrographs, e.g., the Z transform method and
the harmonic analysis method (Dooge, 1973). A previous study (Hanson and Johnson 1964)
classified and compared the usual unit hydrograph calculation methods.
In recent decades, the use of probability distribution functions (pdfs) to develop synthetic unit
hydrographs (SUH) has received much attention because of its similar properties to unit
hydrographs. First, the type of unit hydrograph needs to be subjected to mathematical analysis.
Typical functions, such as a parabola P-III (Yuan , et al., 1991) (Zhai and Li, 2004)curve, can be
used to describe the unit line and a mathematical model of the unit hydrograph can be established.
A previous study (Bhunya, et al., 2007) explored the potential of using four popular pdfs, i.e.,
two-parameter Gamma, three-parameter Beta, two-parameter Weibull, and one-parameter
Chi-square distribution, for deriving a SUH. The Gamma functions are the most widely used
functions (Singh, 2005, 2009; Bhunya, et al., 2003). This approach aims to determine the
relationship between each variable in a unit hydrograph, which facilitates a more in-depth analysis
of a unit hydrograph.
In the present study, we used Gamma functions to describe a unit hydrograph and determined
why a unit hydrograph may follow this distribution. The Gamma function parameters were




optimized using a genetic algorithm. Finally, the unit hydrograph was obtained using the Collins
iterative method.
For a specific basin, the confluence time of a flood is relatively stable and can be determined
according to the flood datas. Therefore, the use of gamma function to derive unit hydrograph is
only in the trial calculation of parameters $\beta$ and $k$. The unit hydrograph expressed by gamma
function based on the combination of these parameters is optimal, which is compared with the use
of P-III function to adjust its statistical parameters. The principle of hydrological frequency
calculation by line fitness of numbers (mean, variation coefficient Cv and skewness coefficient Cs)
is very similar. Meanwhile, the time interval t, parameters and values of gamma function can
parameterize the regional characteristics of river basin confluence, which hasthe important
significance for this study.

## 109    1    Mathematical model and method

### 110    1.1    Mathematical model

In spatial or temporal physical entropy-based modeling of hydrology and water resources, the
cumulative distribution function (CDF) of a design variable (e.g., a flux or a discharge) is
analyzed in terms of its concentration (e.g., stage of flow) (Cui, et al., 2012). The theory of
composition proposed by Zhang (2003) provides a model and a uniform calculation method for
studying the composition of things. This theory considers the analysis of three concepts, i.e., the
general set, the distribution function, and the degree of complexity. This theory is also considered
the most highly approved principle followed by random systems, i.e., the entropyprinciple.
The variable $x$ is continuous and random, and can be viewed as a general set of flag
variables. If the pdf $f(x)$ of $x$ agrees with the followingfunction:

$$f(x) = \frac{\beta^{k}}{(k-1)!} x^{k-1} e^{-\beta x}, x > 0$$

120                                                                (3)

Then the pdf follows a Gamma distribution, where $\beta$ and $k$ are shape and scaleparameters.
This is one of the famous Pearson pdfs, which is known as a Pearson type III distribution. The
curve has a peak with a left-right asymmetry. In nature, many phenomena follow this distribution.
In China, hydrological studies often use the Pearson type III distribution to simulate hydrological
data series, because it has a greater than or equal to zero lower bound on the variable requirements
and its elasticity is greater than the normal distribution (Ye and Xia, 2002). This choice isbased
on experience, but it lacks a theoretical justification.
Using entropy theory, a previous study (Zhang, 2003) described the physical form of this
distribution. By analyzing the structure of Eqn. (3), is not difficult to show that it has a negative
exponential distribution, which is a part of the exponential function, and it also has the
characteristics of a power function in a Pareto-family distribution. The exponential distribution





corresponds to the constraints on the invariant algebraic average of the flag variables, while the
power function corresponds to the constraints on the invariant geometric mean. It may be
speculated that the constraints on the Gamma distribution are the fixed algebraic average and
geometric means of the variables.
In this study, $f(x)$ is the pdf of a positively defined random variable, i.e.,
$$1 = \int_0^\infty f(x) \qquad (4)$$

$u$ represents the algebraic average of variables, thus
$$u = \int_0^\infty x f(x) dx \qquad (5)$$

while $v$ is the geometric mean of the random variable $x$, which can be expressed as the
algebraic average of the logarithm, i.e.,
$$v = \int_0^\infty \ln x f(x) dx \qquad (6)$$

The entropy of the random variable $x$ can be written as
$$H = -\int_0^\infty f(x) \ln x f(x) dx \qquad (7) .$$

Given the constraints in Eqns. (4), (5), and (6), the Lagrange method can be used to estimate
the distribution function $F$ based on the maximum entropy to determine the distribution
function. Thus, $F$ is defined as follows:
$$F = -\int_0^\infty f \ln f dx + C_1(\int_0^\infty f dx - 1) + C_2(\int_0^\infty x f dx - u) + C_3(\int_0^\infty \ln x f dx - v) \qquad (8)$$

Where, $C_1$, $C_2$, and $C_3$ are undetermined constants. The entropy principle demands that the
value of $F$ is maximal. The partial derivative of $f(\bullet)$, i.e., the partial derivative is 0, can be
obtained using Eqn (8). The results are as follows.
$$f(x) = \exp(C_1 - 1 + C_2 x + C_3 \ln x) \qquad (9)$$

This formula can be used to obtain the distribution function. It is the product of the power
function and the exponential function, and its form is identical to a Gamma function.
Hydrological data are random variables that exceed zero. If the hydrological processes are
stationary, then the algebraic average and geometric mean of the hydrological characteristics
variable can be approximated as a fixed value. For example, a mean basin annual runoff is
basically stable (the algebraic average is constant), so the probability of a major flood occurring is





small, and most floods are close to the normal value of the accumulated years (the geometric mean
is constant). However, the uncertainty of a different type of flood occurring each time is maximal.
Thus, the complexity of the outcome is maximized. This is consistent with the following: "In a
generalized (objective, system, sampling experiment), if the algebraic average and geometric
mean of the variables (values of statistical indicators) are constant and the complexity is maximal,
then we can conclude that the probability (the percentage) of the flag value (all the values of the
variables) for each individual must obey a Gamma distribution (Pearson type III distribution)
(Zhang, 2003)."
At the same time, a lot of practical experiences showed that Gamma function can reflect the
characteristics of flood probability. In the view of this, the unit line $q_i$ is defined as Gamma
function.

$$q_i = \frac{\beta^k i^{k-1} e^{-\beta i}}{(k-1)!}, i = 1,2, \quad ,n \qquad (10)$$


When the parameters of $\beta$ and $k$ value were different types, the line type of $q_i$ $q_i$ was
different, as shown in Fig. 1.

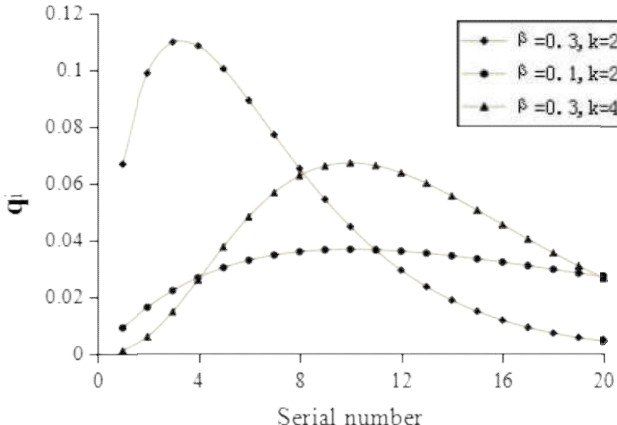


**Fig. 1** The change of function under different parameters

## 175 1.2 Method

### 176 *1.2.1 Genetic algorithms*

Genetic Algorithms (GAs) is an effective global search method, which simulates natural
selection and genetic mechanism. This method of searching the optimal solution of the problem
through natural evolutionary process has been applied in many problems such as function



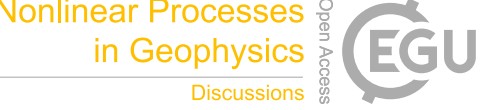

optimization and combinatorial optimization. Genetic algorithm can automatically acquire and
accumulate the knowledge of search space in the search process, and adaptively control the  search
process to find the  best solution(Davis,1991;Michalewicz,1996). The  genetic algorithm regards a
family of  randomly generated  feasible solutions as the   parent population, takes  fitness function
(objective function or one of its transformation forms) as the measurement of the ability of the
parent individual to adapt to the environment, generates the offspring individual through  selection
and  hybridization, and  then mutates the  latter, eliminating the  fittest and  the fittest, so that the
individual adapts to the environment through repeated evolutionary iterations. With the continuous
improvement of  ability,  excellent individuals keep  approaching the  optimum point(Yuan,2002).
After several generations, the algorithm converges to the best individual. The best individual in a
group is likely to be the optimal or approximate optimal solution of the problem.
As a new random search and optimization method to simulate biological evolution, genetic
algorithm has been widely used  in the  field of  optimization(Chen,1996;Li,2009). The  parameter
optimization of many empirical formulas of hydrological models is essentially based on the  global
optimization ability of genetic algorithm.

### 195  *1.2.2  Collins iteration method*

Iterative method is a mathematical process to solve  the  problem by finding  approximate
solutions that meet the restrictive conditions from an  initial value. Iterative algorithm is also a
basic method to solve problems by computer. It makes use of the characteristic of fast computing
speed and suitable for repetitive operation, so that the computer can repeat a set of instructions  (or
steps). When the instructions (or steps) are executed, a new value of the variable will be derived
from the original value of the variable. Assuming that we  want to derive an approximate solution,
we should determine an initial value, an iteration function and a restriction condition according to
the actual situation and data firstly, until the absolute value of the initial value and the calculated
approximation value is less than a certain value. That is to say, we find the exact desired value.

### 205  2    Approach used to determined the unit hydrograph

The overall calculation  process is divided  into two parts: (1) the parameters  of the unit
hydrograph are optimized using the genetic algorithm, so the initial unit hydrograph  can be
calculated; and (2) the final unit hydrograph is calculated using the Collins iterative method.

### 209  2.1  Calculation of the initial unit hydrograph using a genetic algorithm

(a) Parametrization of Gamma Function
For simplicity, Eqn. (10) is transformed as follows
$$
\begin{cases}
q_i = \dfrac{1}{(x_1 - 1)} x \cdot x_2 (i + x_3)^{(x_2-1)} \cdot e^{-x_1 \cdot (i + x_3)}, \; i = 2, \quad , n\text{-}1 \\
q_1 = q_2 = 0 \\
\quad \quad \quad \quad n
\end{cases}
\tag{11}
$$


Where, $x_1$, $x_2$, and $x_3$ are the constants of the unit hydrograph $q_i$, which is the argument
of the problem for the genetic algorithm. $i$ is the argument of the unit hydrograph, which is a
period number. $i$ is the known number, which is processed by the genetic algorithm. n is the
time period of the unit hydrograph, which is defined according to the actual engineering problem.
The variables $x_1$, $x_2$, and $x_3$ are in a range of [0, 5]. The chromosome is coded as a floating
point value.
If one chromosome is $v' =$ [1.7450, 8.7014, 1.5042] and $n = 11$ (n is the time period of
the unit hydrograph), the unit hydrograph obtained using the constant given above is as follows.

$$
\begin{cases}
q_i = \cfrac{\cdot\,1.7450^{8.7014}}{(8.7014\text{-}1)}\cdot(i+1.5042)^{(8.7014-1)}\cdot e^{-1.7450(i+1.5042)}, \\
\qquad\qquad\qquad\qquad\qquad\qquad\qquad i = 2,3,\ \ ,10 \\
q_1 = q_{11} = 0
\end{cases}
$$

(12)

Using Eqn. (12), the results obtained for the unit hydrograph are [0, 570, 688, 563, 356, 187, 223
        86, 35, 13, 5, 0], as shown in Fig. 2.

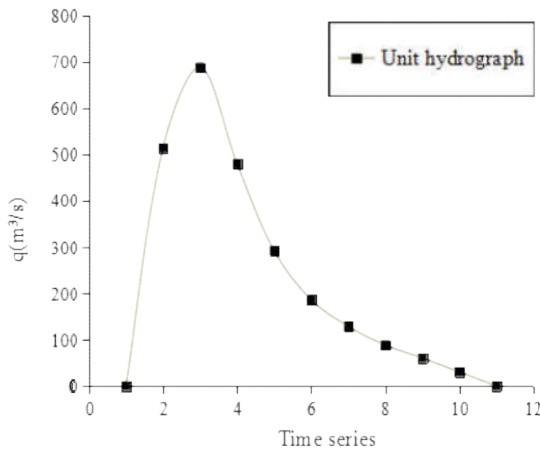


**Fig. 2**   Unit hydrograph $q_i'$

(b) Determination of objective function
According to the basic principle used to derive the unit hydrograph, the objective  function of
the genetic algorithm can be expressed as follows:

$$
\max:\ \varphi(q_i(x_1,x_2,x_3)) = \frac{1}{(Q'(q_i)-Q)^2}
$$

(13)



Where $Q'(q_i)$ is the converging flow obtained by the unit hydrograph $q_i(x_1, x_2, x_3)$ in a
basin, and $Q$ is the measured flow in the basin outlet section.
Physical interpretation of the objective function Eqn. (13)
A set of parameters, $x_1^*, x_2^*$, and $x_3^*$, are searched based on the following conditions. The
inverse square of the difference between $Q'(q_i)$ and $Q$ is maximal. To avoid computations if
the objective function value is too small, the expansion coefficient $M$ is introduced. The value
of $M$ is determined according to the specific situation. Eqn. (13) is converted into the following
form.

$$\max : \varphi(q_i(x_1, x_2, x_3)) = \frac{M}{(Q'(q_i) - Q)^2} \quad (14)$$


(c) Optimization parameters
We use the steps shown in Figure 3 of genetic algorithm to optimize the unit lineparameters:

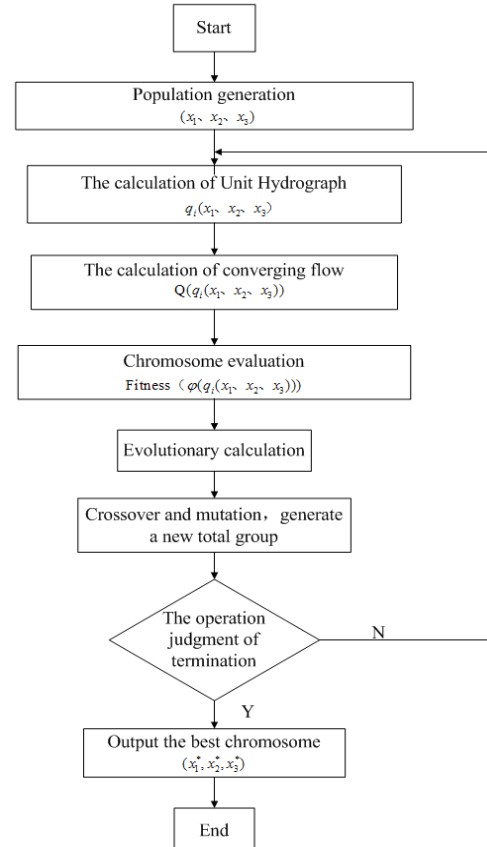





**Fig. 3**  The processes map that was used the genetic algorithm to optimize the parameters
The optimum parameters $(x_1^*, x_2^*, x_3^*)$ are obtained by the above steps, and then the unit line
is calculated by the optimum parameters, as follows:
$$\begin{cases} q_i = \dfrac{x_2^*}{(x_2^*-1)!} \cdot (i+x_3^*)^{(x_2^*-1)} \cdot e^{-x_1^* \cdot (i+x_3^*)} , i = 2, \quad , n\text{ - }1 \\ q_1 = q_n = 0 \end{cases} \quad (15)$$

### 2.2  Calculation of the final unit hydrograph using the iterative method

Collins iteration method is used to calculate the final unit hydrograph. Firstly, each period of
the net rainfall runoff process is calculated by unit hydrograph $q^*$. Meanwhile, the maximum net
rainfall $h_{max}$ and its runoff process $Q(q_i^*, h_{max})$ are determined, and overall net rainfall total
runoff $\sum Q(q_i^*, \overline{h_{max}})$ is calculated; Secondly, another unit hydrograph
$q_i' = \dfrac{Q - \sum Q(q_i^*, h_{max})}{h_{max}}$ is deduced, and new $q_i'$ is deduced continuously by $q_i^*$ according to
the restriction condition of $\varepsilon = \left| q_i^* - q_i' \right| \le ErrorExcepted$ until the error between the two
units meets the requirement, then the final unit hydrograph $q_i = q_i'$.

## 3    Examples

**Example 1**

In Table 1, the data were taken from a previous study (Zhuang and Lin, 1986).
**Table 1**   The calculations of example 1
R: unit hydrograph; Q: measured discharge; $Q'$、$q'$: discharge and unit hydrograph of trial and error method;
discharge and unit hygrograph of GACIM

| Time series(h) | R(mm) | The measured runoff Q(m³/s) | The trial and error method | | GACIM | |
|---|---|---|---|---|---|---|
| | | | $Q'$ (m³/s) | $q'$ (m³/s) | $Q''$ (m³/s) | $q''$ (m³/s) |
| ① | ② | ③ | ④ | ⑤ | ⑥ | ⑦ |
| 0 | | 0 | 0 | 0 | 0 | 0 |
| 6 | 3.8 | 0 | 0 | 0 | 0 | 0 |
| 12 | 3.9 | 50 | 190 | 500 | 195 | 514 |
| 18 | 0 | 252 | 455 | 685 | 461 | 687 |
| 24 | 27.3 | 662 | 446 | 470 | 450 | 480 |
| 30 | 2.9 | 1700 | 1650 | 280 | 1700 | 292 |





| | | | | | |
|---|---|---|---|---|---|
| 36 | 2210 | 2200 | 195 | 2210 | 186 |
| 42 | 1630 | 1610 | 125 | 1630 | 129 |
| 48 | 1020 | 981 | 85 | 1020 | 88 |
| 54 | 650 | 669 | 60 | 650 | 60 |
| 60 | 440 | 433 | 35 | 440 | 30 |
| 66 | 290 | 288 | 15 | 290 | 0 |
| 72 | 190 | 195 | 0 | 190 | |
| 78 | 100 | 113 | | 100 | |
| 84 | 40 | 51 | | 9 | |
| 90 | 0 | 4 | | 0 | |


The unit hydrographs determined using the two methods are shown in columns (5) and (7) in
Table 1. A comparison of the unit hydrographs is shown in Figure 4. The flow processes
calculated using the unit hydrographs are shown in columns (4) and (6) in Table 1. A comparison
between the calculated flow process and the measured flow is shown in Figure 5.

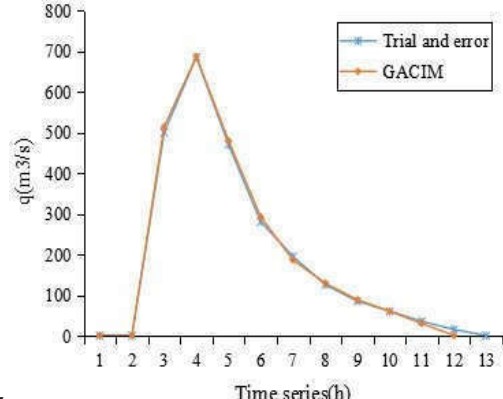


**Fig. 4**   Example1: Unit hydrographs ascertained by two methods


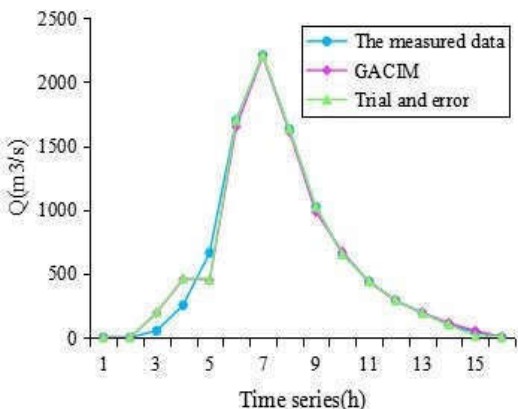


**Fig. 5** Example1: The flow process of outlet section
The actual hydrological data in Table 1 show that the period number for the runoff was 5 and
the period number for the flow process in the outlet section was 15. According to the theory of the
unit hydrograph, the period number for the unit hydrograph should be $15 - 5 + 1 = 11$. Using the
GACIM (genetic algorithm and the Collins iterative method), the period number for the unit
hydrograph was 11. Using the trial and error method, the period number for the unit hydrograph
was 12. To further consider the performance of the two methods, we compared the flow process in
the outlet section and the results obtained using the two unit hydrographs (measured value and
calculated value). A statistical analysis of the results is shown in Table 2.
**Table 2** The error statistics of example 1

| Project Method | GACIM | The trial and error |
|---|---|---|
| The error of flood peak($m^3$/s) | 0 | 10 |
| The maximum error of discharge($m^3$/s) | 212 | 216 |
| The average absolute error of discharge($m^3$/s) | 37.31 | 46.19 |
| The total error of flood peak discharge($m^3$/s.h) | $-111$ | $-51$ |
| The relative error of flood peak discharge (%) | $-1.20$ | $-0.55$ |


### 279 Example 2

In Table 3, the data were taken from a previous study (Li and Zheng, 1982).
**Table 3** The calculations of example 2

| Time series(h) | R(mm) | The measured runoff Q($m^3$/s) | The trial and error method | GACIM |
|---|---|---|---|---|




| ① | ② | ③ | $Q^{'}$ (m³/s) ④ | $q^{'}$ (m³/s) ⑤ | $Q^{''}$ (m³/s) ⑥ | $q^{''}$ (m³/s) ⑦ |
|---|---|---|---|---|---|---|
| 0 |  | 0 | 0 | 0 | 0 | 0 |
| 6 | 15.3 | 97 | 96 | 63 | 97 | 49 |
| 12 | 7.4 | 214 | 215 | 110 | 214 | 119 |
| 18 | 5.8 | 304 | 308 | 124 | 304 | 149 |
| 24 |  | 371 | 374 | 143 | 371 | 128 |
| 30 |  | 294 | 294 | 76 | 294 | 85 |
| 36 |  | 190 | 202 | 41 | 190 | 47 |
| 42 |  | 123 | 120 | 30 | 123 | 23 |
| 48 |  | 80 | 80 | 22 | 80 | 10 |
| 54 |  | 49 | 52 | 12 | 49 | 4 |
| 60 |  | 30 | 22 | 0 | 19 | 0 |
| 66 |  | 15 | 7 |  | 4 |  |
| 72 |  | 0 | 0 |  | 0 |  |


The unit hydrographs determined using the two methods are shown in columns (5) and (7) in
Table 3. A comparison of the unit hydrograph is shown in Figure 6. The flow processes calculated
using the unit hydrographs are shown in columns (4) and (6) in Table 3. A comparison of the
calculated flow process and the measured flow is shown in Figure 7.

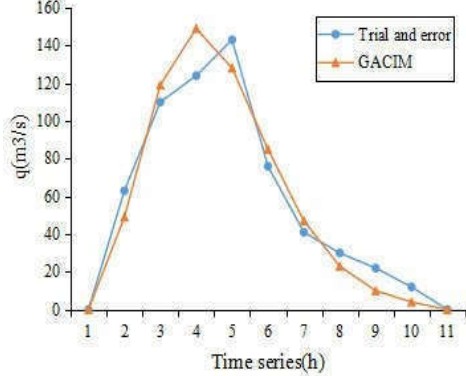


**Fig. 6**   Example2: Unit hydrographs ascertained by two methods



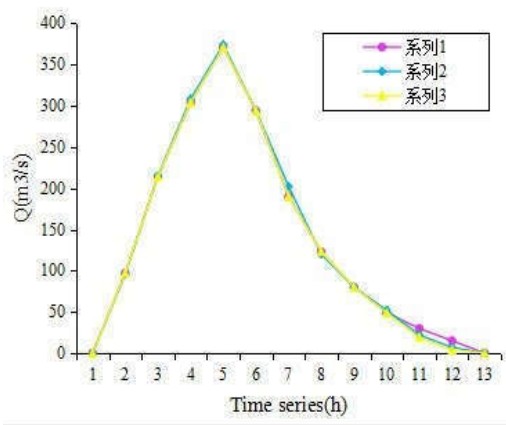

**Fig. 7**   Example2: The flow process of outletsection

Figure 4 shows that GACIM was significantly   better than the trial and error method in terms

of the shape of the curve. We also compared the flow process in the outlet section and the data
obtained using the two unit hydrographs. A statistical analysis of the results is shown in Table4.

**Table 4**    The error statistics of example2

| Project Method | GACIM | The trial and error |
|---|---|---|
| The error of flood peak(m3/s) | 0 | -3 |
| The maximum error of discharge(m3/s) | 11 | -12 |
| The average absolute error of discharge(m3/s) | 1.69 | -0.23 |
| The total error of   flood peak discharge(m3/s.h) | 22 | $-3$ |
| The relative error of flood peak discharge (%) | 1.25 | -0.17 |


From Table 2 and Table 4, the calculation accuracy of BGACM is obviously better than

that of trial-and-error method in most projects. Although the total error of flood volume is larger
than that of trial-and-error method, the relative error of flood volume is only 1.2% and 1.25%, so it
does not affect the application of actual projects.

Figure 6 and 7 show that the flow processes of the two unit hydrographs weresimilar.

However, a comparison of the shapes of the unit hydrograph showed that the continuity  and
smoothness of GACIM were better than the trial and error method. The GACIM method
conformed better with the features of a time-invariantsystem.

It can be seen that BGACM  method is  better at simulating river basin confluence  process,

which depends on the physical mechanism of the algorithm, while trial-and-error method pays
more attention to the balance of total flood volume. This is the respectivecharacteristics and
advantages of the two algorithms exactly.



## 4    Discussion

The present study used a combination of a genetic algorithm and the Collins iterative method (GACIM) for determining a unit hydrograph. The method and implementation steps were described, while examples and analyses were used to demonstrate the scientificity, reliability and practicability of this method. The outcomes of this study are discussed below.

(a) Difference between GACIM and other methods

In principle, GACIM is based on composition theory and it describes the physical mechanism and process of flood confluence using mathematical equations. Using the basic concept of the unit hydrograph and a genetic algorithm as a mathematical tool, this method can be used to simulate the flood confluence process.

Therefore, the simulation of the convergence process is more accurate with GACIM.

Other methods for calculating unit hydrographs include the analysis method, least squares method, and the trial and error method. These methods are more focused on the unit hydrograph as an outlet flow process and they fit the measured flow precisely, but they ignore the composition and structure of the unit hydrograph itself. Example 2 shows that GACIM performed better at simulating the basin confluence process, whereas other methods paid more attention to the balance of the total flood volume.

(b) Genetic operator design issues

A genetic algorithm is a very useful optimization tool. Its biggest advantage is that it has wide adaptability and unlimited problem space, so it can handle many different constraints. This strategy uses a penalty factor. This is because the genetic algorithm method delivers exhaustive engineering accuracy if the population is sufficiently large.

There are two types of genetic algorithm, i.e., the standard genetic algorithm (crossover and mutation) and evolutionary computing (selection). A genetic algorithm simulates the recombination of genes to create new offspring in each generation, whereas evolutionary computation is a population process that updates each generation.

In this study, a genetic algorithm was used to optimize the parameters of the Gamma function and the unit hydrograph was calculated according to the law of basin confluence. Thus, the parameters were generated by a genetic algorithm. Therefore, the design of the genetic operators is related directly to whether reasonable generation parameters could be obtained.

A genetic algorithm has two components: crossover and mutation. Crossover is the main genetic operation that generates new individuals, but it also maintains the relative stability of the population at the same time. However, the variation is a basic calculation and the main effect is to produce a new gene from the population, which provides new information for the population.

In general, the initial population of the genetic algorithm is generated in the value space. Crossover and mutation are performed in the value space. In the present study, the value of the Gamma function was in a certain range. Initially, we could not define a reasonable space. If the value space is too large, a bigger population must be used to meet the needs of the individual

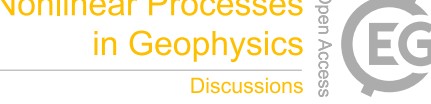



distribution density. However, this greatly reduces the computational speed. If the value space is
too small, it might not meet the parameter combination required for the engineering precision. To
solve this problem, we observed the following principles during the design of the genetic operator.
(i) The crossover operator was determined where a new individual was generated at random
in the space [0,a] (a>0).
(ii) The random expansion value space of the mutation operator was [0 a] and its amplitude
was random. For the floating point coding mutation operator, the following program was
implemented in MATLAB:
1) numMut=round(size(parent,1)*Ops/2); to calculate the number of variations
2) numPop=size(parent,1); to calculate the size of the population
3) numPara=size(parent,2); to calculate the number of parameters
4) for j=1:numPara
5) for i=1:numMut
6) a=round(rand*(numPop-2)+2); select a male parent
7) parent(a,numPara) = parent(a,numPara)*(1+rand/gen); generate a new generation
8) end
9) end.
The parameters of the offspring chromosome were calculated during step 7) of this program,
where the variation in the amplitude was related to the number of evolutionary passages. The
variation in the amplitude declined gradually with increasing passage numbers.
Figure 8 illustrates the crossover operator and mutation operator with the passage of
evolutionary time.

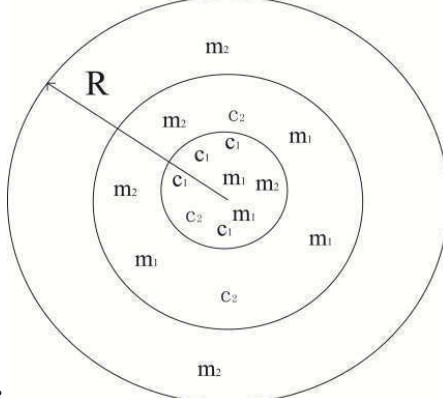

**Fig. 8** The sketch map of the continuation of parameters value space

In the first generation, the filial generation caused by the crossover operator was still in the
initial parameter space, which corresponds to the inner loop in Figure 8. However, the filial
generation caused by the mutation operator was beyond this range and it expanded to the second

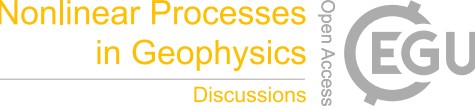

ring.  In the  second generation, the  filial generation of the crossover operator was  extended to the
second ring while the filial generation of the mutation operator was extended to the third ring.  The
expanding amplitude of  the  adjacent ring  decreased with  the  passage of  evolutionary time. This
method was repeated until the predetermined evolutionary algebra was completed.

In Examples 1 and 2, the  initial values of the parameters were [0 5]. The two sets of

optimized parameters were as follows.
Example 1: [1.7450, 8.7014, 1.5042]

Example 2: [1.6052, 7.9209, 0.30007]

These two examples demonstrate the design rationality  and the  validity of the  genetic

operator.

(c) Research methods for hydrological analysis and calculation

The factors that affect hydrological phenomena are  very  complex. There is  still no  accurate

understanding of the  causal  relationships among hydrologic phenomena. It is  considered that
hydrological phenomena involve certainty and randomness, which form  the basis  of  hydrological
research. Therefore, causal analysis  and  probabilistic  statistics  are  the  main  methods used  for
hydrological analysis and calculation. In  practical applications, causal analysis is confined mostly
to  qualitative analysis.  Quantitative problems demand empirical statistical relationships based  on
actual observational data.

Based on the theory of composition, the distribution function in statistical physics has been

extended to hydrology as  a non-physics  field.  Thus,  hydrological systems  can be  viewed as  a
generalized  collection. The regularities of hydrological phenomena  have  been  simulated using
distribution  functions. Distribution  functions and  functional relationships have  been  determined
using observation data, which generally means that objective laws are formalized.

The present study was a preliminary attempt to  investigate the  quantitative relationships

among  hydrological phenomena based on the  theory of composition and its distribution  function.
The author believes that this theory could be a new approach to exploring hydrological rules.
**Author Contributions:** Hongyan Li conceived and designed the paper; Cidan Yangzong analyzed the
data; Hongyan Li and Cidan Yangzong wrote the paper. wrote the paper.
**Foundation item:** The author hereby would like to express deep gratitude to the key special project of
"Efficient 399 Development and Utilization of Water Resources" (2017YFC0406005), China-ROK
cooperation project (51711540299), 400 Natural Science Foundation of Jilin Province (20180101078JC)
and other projects which have given support to the 401 research of this paper.
**Acknowledgments:** The authors hereby would like to express deep gratitude to the key special project
of "Efficient Development and Utilization of Water Resources" (2017YFC0406005), China-ROK
cooperation project (51711540299), Natural Science Foundation of Jilin Province (20180101078JC)
and other projects which have given support to the research of this paper.
**Data Availability Statement:** The data sources and code used are provided in the article





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
