# Peer review of "Application of Gamma functions to the determination of"

_Nonlinear Processes in Geophysics, 2020_

## Referee Comment (RC1) · Anonymous Referee #1 · 5 May 2020

Dear authors,

I applaud for your effort on the contribution, however I do not think that the overall manuscript is novel enough especially in hydrological science. In addition, the manuscript characteristics also lean towards 'technical note' instead of 'research article' status. I do find the writing style to be rather clear and not complicated.

From my understanding the paper focuses on the parameter optimisation of classic Collin's method in derriving UH. The author chose genetic algorithm for this optimisation purpose. Both of these methods are very well documented and long known in Hydrology, therefore the manuscript does not present anything novel.

Meanwhile I also think that the writing or presentation is not adequate, it contains a

[Figure]

<a href="#">Printer-friendly version</a>

<a href="#">Discussion paper</a>

lot of typos, content is not concise, bad quality figures and equations, and non-critical result presentations.

Herein, I have also attached the commented manuscript with few highlights of the mentioned. I would encourage the authors to try submitting to some technical note journals upon having the manuscript revised with a more concise and critical presentation.

Good luck.

Please also note the supplement to this comment:
https://www.nonlin-processes-geophys-discuss.net/npg-2020-1/npg-2020-1-RC1-supplement.pdf

**Supplement:**

[revised manuscript text omitted]

---

## Referee Comment (RC2) · Anonymous Referee #2 · 12 Jun 2020

My recommendation is in the same line as Referee #1: encourage the authors to try submitting to some technical note journals upon having the manuscript revised with a more concise and critical presentation.

The reasons for this recommendation are the following: (i) The unit hydrograph is an idealized linear concept, whereas this Journal is about nonlinear processes in Geophysics, as clear from the name of the Journal. The possible non-linearity of the method of fitting does not make the paper suitable for the Journal either, because a fitting method is not a Geophysics process. (ii) In the article, there is no consideration of the unit hydrograph concept concerning its limitations or possible generalization to less idealized and possible nonlinear ideas to model the hydrologic response. For instance, the hypothesis of uniform rainfall is far from the current understanding of the

irregularity of rainfall fields. The rain is just one example of a nontrivial, highly nonlinear process not taken into account. (iii) The quality of the paper concerning writing, figures, and equations needs substantial improvement (iv) The substance of the paper, a fitting method is not tested adequately according to standard modern techniques.

---

## Author Comment (AC1) · 12 Aug 2020

Thank you for your evaluation and careful revision of the article, I will treat it with heart. According to what you said, the parameter optimization of Collin iteration method and genetic algorithm is more common in hydrological research, but there are few applications in unit line estimation. There are many methods for deriving the unit line, but these methods have certain requirements for flood data when deriving the unit line and the estimated unit line may not be optimal. Therefore, this paper uses the Gamma function to simulate the law of the basin confluence process, and at the same time, the Gamma function parameters And based on the condition that the error between the calculation process and the measured process is minimum, the genetic algorithm is used to optimize the parameters of the Gamma function, the initial unit line is derived,

and then the final unit line is calculated using the Collins iterative method. The comparison and analysis of the results of the actual calculation examples show that the unit line deduced by this method has better accuracy than the general method, and at the same time, it can reveal the watershed confluence law. The main research purpose of this paper is to apply newer methods commonly used to optimize and improve the accuracy of mature methods.

---

## Author Comment (AC2) · 12 Aug 2020

First of all thank you for your careful review of this article. (i) This article mainly studies the improvement process of the unit line estimation method. The unit curve is very common in hydrology, and there are many methods for its estimation. However, these methods not only have limitations, but also have poor results. Therefore, it is necessary to introduce more novel methods to improve the estimation results and methods of the unit curve. (ii) In this paper, the unit line estimation equation is expressed as a non-linear gamma function and parameterized, and the unit line estimation result is improved by the method of parameter optimization by genetic algorithm. (iii) The limitations of the previously commonly used unit line estimation methods are explained in the form of the applicability conditions of the methods in the introduction of these

methods.